# Response Surface Methodology to Efficiently Optimize Intracellular Delivery by Photoporation

**DOI:** 10.3390/ijms24043147

**Published:** 2023-02-05

**Authors:** Ilia Goemaere, Deep Punj, Aranit Harizaj, Jessica Woolston, Sofie Thys, Karen Sterck, Stefaan C. De Smedt, Winnok H. De Vos, Kevin Braeckmans

**Affiliations:** 1Laboratory of General Biochemistry and Physical Pharmacy, Faculty of Pharmaceutical Sciences, Ghent University, Ottergemsesteenweg 460, 9000 Ghent, Belgium; 2Laboratory of Cell Biology and Histology, Department of Veterinary Sciences, University of Antwerp, Universiteitsplein 1, 2610 Antwerp, Belgium

**Keywords:** response surface methodology, polydopamine nanoparticles, photoporation, intracellular delivery, cell therapy

## Abstract

Photoporation is an up-and-coming technology for the gentle and efficient transfection of cells. Inherent to the application of photoporation is the optimization of several process parameters, such as laser fluence and sensitizing particle concentration, which is typically done one factor at a time (OFAT). However, this approach is tedious and runs the risk of missing a global optimum. Therefore, in this study, we explored whether response surface methodology (RSM) would allow for more efficient optimization of the photoporation procedure. As a case study, FITC-dextran molecules of 500 kDa were delivered to RAW264.7 mouse macrophage-like cells, making use of polydopamine nanoparticles (PDNPs) as photoporation sensitizers. Parameters that were varied to obtain an optimal delivery yield were PDNP size, PDNP concentration and laser fluence. Two established RSM designs were compared: the central composite design and the Box-Behnken design. Model fitting was followed by statistical assessment, validation, and response surface analysis. Both designs successfully identified a delivery yield optimum five- to eight-fold more efficiently than when using OFAT methodology while revealing a strong dependence on PDNP size within the design space. In conclusion, RSM proves to be a valuable approach to efficiently optimize photoporation conditions for a particular cell type.

## 1. Introduction

Instrumental to understanding and harnessing the functioning of biological systems at their most fundamental levels is the intracellular delivery of molecules (e.g., small biomolecules, nucleic acids, proteins and synthetic nanomaterials) that do not passively diffuse across the cell membrane under normal circumstances [1,2,3]. To this end, a plethora of intracellular delivery methods has been developed, which can be categorized as either carrier-mediated [4,5,6,7] or membrane-disruption-mediated methods [8]. Though carrier-mediated delivery shows great therapeutic potential, particularly for in vivo applications, interest in physical membrane-disruption-mediated delivery techniques has risen over the past few years, especially for in vitro or ex vivo manipulation of cells. Here, the cell membrane is transiently perturbed by physical phenomena (e.g., of mechanical, optical, thermal, magnetic and/or electrical nature), creating membrane pores through which molecules of interest can enter the cells [2,8]. Examples of physical membrane disruption techniques are electroporation [9], sonoporation [10], magnetofection [2,11] and microfluidic mechanoporation [12].

Emerging as another capable and flexible physical intracellular delivery method is photoporation. In photoporation, a laser irradiates photothermal nanoparticles (NPs), which are highly efficient in converting electromagnetic energy (i.e., light) into thermal energy [13,14]. When the photothermal NPs are present at or close to the cell membrane, laser irradiation induces the formation of pores by means of thermal, mechanical and/or chemical effects [14]. When the NP its temperature increases above the critical temperature of the water, a distinct mechanical effect can arise, which is the generation of water vapor nanobubbles (VNBs) [15]. VNB-formation takes place when the heated NPs evaporate the liquid in their immediate vicinity. The VNB will increase in size until the NPs’ thermal energy is depleted, at which point the VNB collapses again [13,16,17,18]. Rapid expansion and collapse of VNBs induce mechanical stress on the nearby cell membrane, resulting in pore formation [19,20]. So far, mostly metallic (e.g., gold and iron oxide) [21,22,23,24,25,26,27,28] or carbon-based (e.g., carbon black, reduced graphene oxide and graphene quantum dots) [29,30,31,32,33,34,35,36] NPs have been used for the photoporation of cells. However, concerns regarding the potentially toxic effects of such essentially non-degradable nanoparticles [37,38,39,40,41,42,43] have led researchers to seek more biocompatible alternatives, such as polydopamine NPs (PDNPs). PDNPs possess excellent photothermal properties across the visible range of wavelengths [44,45,46,47]. Moreover, a broad range of PDNP sizes can be readily synthesized in an ecological manner, and their highly functionalized surface allows for chemical modifications in a multitude of ways [48,49,50,51,52]. Successful use of PDNPs in the photoporation of human T cells for the delivery of mRNA shows a promising future for these NPs [53].

When using photoporation to deliver cargo molecules into cells, several parameters need to be optimized to maximize the delivery efficiency while minimizing cytotoxicity. Important experimental factors include NP size [53], NP concentration [24,25] and laser fluence [19]. In photoporation studies so far, these factors are optimized with a one-factor-at-a-time (OFAT) approach [54]. While OFAT is commonly used in experimental sciences, it is time- and resource-consuming and runs the risk of not finding the true global maximum [54,55]. This can be remedied by using response surface methodology (RSM), which was first introduced by Box and Wilson in 1951 [56]. Derived from the design of experiments, RSM consists of a collection of statistical and mathematical tools, which allow a researcher to gain knowledge of a certain process and optimize it as efficiently as possible, not requiring replicates [57,58]. This can be done by careful selection of a limited set of data points to be collected, termed *the experimental design*, and fitting a low-order polynomial to the results using ordinary least squares (OLS) regression [59,60,61,62,63].

Currently, the applicability of RSM for the optimization of photoporation parameters has not been explored. Therefore, we investigated this for several parameters of interest: PDNP size, PDNP concentration and laser fluence. With each having their particular merits, two commonly used RSM designs were compared, a central composite design and a Box-Behnken design, to determine the parameter values that lead to an optimal delivery yield of FITC-dextran 500 kDa (FD500) cargo molecules in RAW264.7 murine macrophage-like cells. Fitted models were statistically evaluated and experimentally validated. Insights from the models were related to the physicochemical characteristics of the used PDNPs, showing that RSM can provide insights beyond mere process optimization.

## 2. Results

### 2.1. Synthesis of Polydopamine Nanoparticles

The first parameter that can influence the outcome of photoporation experiments is the size of the PDNPs. As summarized in Section 4.8.1, the following sizes were included in the experimental design: 142 nm, 300 nm, 500 nm, 700 nm and 858 nm. Using 3.5 mg/mL dopamine (DA), a DA:NaOH ratio of 1:0.7 and a reaction temperature of ca. 50 °C resulted in a gradual growth of PDNPs, as evidenced by DLS size measurements at regular intervals (Figure 1A). This gradual growth allowed for a refined control over the NP size by tuning the reaction time, DA concentration and DA:NaOH ratio. Five distinct sizes of PDNPs were prepared this way, based on having a hydrodynamic diameter close to the desired values of the experimental designs. UV-Vis spectrophotometry showed a clear change in the extinction spectrum between the DA monomer solution and the PDNP dispersions, as exemplified in Figure 1B for 500 nm PDNPs.

To improve colloidal stability and facilitate the interaction of PDNPs with cells, a coating of BSA was additionally applied to the PDNPs, which did not affect the UV-Vis spectrum (Figure 1B). The size of PDNPs was measured in Opti-MEM, as this is the medium that was used for photoporation experiments. In the absence of the BSA coating, agglomeration occurred in Opti-MEM (Figure 1C), which was not observed for the BSA-coated PDNPs (Figure 1C). A summary of the size and polydispersity index (PDI) of all PDNPs used in this study is provided in Table 1 and Appendix A.

Most importantly, PDNPs could be synthesized with a monodisperse size close to what is required in the model design, regardless of incubation in HyClone Pure water or Opti-MEM (Figure 2A–C and Appendix A). All synthesized PDNPs demonstrated excellent long-term colloidal stability during a period of 90 days, considering that their hydrodynamic size and PDI measured by DLS remained virtually unaltered after 90 days (Appendix A). Moreover, it became evident that all PDNPs had a negative zeta potential (Z.P.), as expected after BSA coating [53,64], which helps to explain their colloidal stability [65]. Further characterization of the PDNPs by scanning electron microscopy (SEM) and transmission electron microscopy (TEM) revealed a uniform, quasi-spherical shape (Figure 2D–I) that was observed across the entire used PDNP size range. The PDNP diameter, as determined from SEM and TEM images (Appendix A and Appendix A), was smaller than the hydrodynamic diameter measured by DLS, which is consistent with earlier studies [53,64].

Next, the photothermal behavior of the different PDNPs was assessed by determination of the VNB-threshold–that is, the laser fluence at which 90% of the irradiated NPs form VNBs that are visible in darkfield microscopy images (Figure 3A) [19]. Plotting the normalized number of VNBs as a function of laser fluence and fitting a Boltzmann curve (Figure 3B–F) enables the identification of the VNB-threshold. For the 142 nm PDNPs, no VNBs could be generated within the range of fluences achievable with our optical setup, indicating a PDNP size threshold for VNB formation (Figure 3B). For 300 nm PDNPs, VNBs began to appear (Figure 3C), but it was only from 500 nm PDNPs onwards that the expected sigmoidal curve was obtained, as is needed to reliably determine the VNB threshold (Figure 3D–F). The VNB-thresholds were determined to be 1.33 J/cm^2^, 1.12 J/cm^2^ and 0.95 J/cm^2^ for 500 nm, 700 nm and 858 nm PDNPs, respectively.

### 2.2. Optimization of Delivery Yield by Response Surface Methodology

#### 2.2.1. Model Fitting and Statistical Assessment

Two commonly used RSM designs were created to compare their performance in optimizing photoporation parameters (PDNP size, PDNP concentration and laser pulse fluence). An eighteen-run Circumscribed Central Composite (CCC) design–belonging to the class of Central Composite Designs (CCDs)–was generated as the first design with five-factor levels for each parameter, representing a spherical and flexible design space [66]. The second design was a sixteen-run Box-Behnken design (BBD), only requiring three-factor levels, as a simpler design, which does not require the inclusion of extreme parameter values and simultaneous combinations thereof that can be technically unattainable at times [67,68]. Both designs were repeated three times to achieve an indication of reproducibility [69]. Ordinary Least Squares (OLS) on the experimental results allowed for an analysis of variance (ANOVA). As can be seen from the bottom two rows in Appendix A, all replicates of the CCC design and BBD resulted in significant models that did not show any significant lack-of-fit. Note that the intercept terms were all significant, but they do not hold any information on the contribution of the design factors. Upon investigating the models’ terms, only the size of the PDNPs was consistently coming up as a significant factor in both the linear and quadratic terms, regardless of the design used. In addition, the CCC design-based models identified the quadratic PDNP concentration term (Conc.^2^) and quadratic fluence term (Fluence^2^) to be weakly significant in two of the three replicates, providing no convincing proof of a consistently significant contribution of these variables. Both designs thus led to models that suggest a considerable dependency of the delivery yield on the PDNP size, whereas the PDNP concentration and laser pulse fluence only seem to have relatively minor effects on the investigated response within the defined ranges, with the BBD-based models generally estimating them to be less of importance than the models based on the CCC design.

Next, we investigated how well each model predicted the delivery yields that were included in the model fitting process—i.e., model fit. We found that the BBD performed better in this regard than the CCC design, as concluded from the summary statistics related to the distance between an observed value and the predicted mean value for that data point (Table 2). This can be seen from the BDD-based models’ consistently higher coefficients of determination (multiple R^2^) and adjusted R^2^ (adj. R^2^). Higher values of statistics, such as the root mean squared error (RMSE) and mean absolute error (MAE) for the CCC design-based models, also demonstrate a less adequate fit relative to the BBD-based models. Preliminary metrics of the models’ predictive powers, also based on the fit itself, again suggest the BBD result in better fitting models, as reflected by their seemingly higher predicted R^2^ (pred. R^2^) and lower predicted residual sum of squares (PRESS) values. However, relatively small pred. R^2^ values in contrast to both multiple R^2^ and adj. R^2^, for all cases, signifies potentially tenuous reliability of the models to make predictions in the design space but can only be confirmed by testing the fit on data that was not used to fit the model.

The primary assumptions of a well-fitted regression model are the normal distribution and homoscedasticity of the residuals. None of the replicates of the two designs showed a significant departure from normality in their residuals, as indicated by their Q-Q plots (Appendix A) and non-significant *p*-values (*p* > 0.05) for the Shapiro–Wilk test (Appendix A) and homoscedasticity was confirmed for all models (*p* > 0.05) by the Bruesch-Pagan test (Appendix A). For all experiments, the run order did not have a definite impact on the introduction of bias by a time-effect (Appendix A), as confirmed by a Durbin-Watson test (Appendix A), which revealed no significant autocorrelation (*p* > 0.05). Thus, as no deviations from the model assumptions were present, inferences could be reliably made based on the model outputs and (statistical) observations.

#### 2.2.2. Analysis of the Response Surface

Having verified the models’ validity, the nature of the response surface and stationary points could be evaluated next. The CCC design-based models identified a stationary point that was practically identical between the three replicates (Table 3), with relatively low coefficients of variation (C.V.). Moreover, all three replicates had a set of three negative eigenvalues, revealing that the stationary point is indeed an optimum, as desired in this application. Results for the BBD-based models were more variable, as clearly reflected by their sizeable C.V. While the NP size at the stationary point is rather similar for all replicates, both the fluence and NP concentration at the stationary point differ from replicate to replicate. Except for the first replicate, which shows a clear optimum, the canonical analysis revealed that some eigenvalues were close to zero (i.e., ridge system) and had mixed signs (i.e., saddle point), thus implying that there was no clear (unique) optimum within the explored design space and largely explaining the variation between the replicate outputs.

By keeping one of the three factors constant at its stationary point value and plotting the surface for the other two factors with the delivery yield as a response, the difference in the nature of the BBD-based and CCC design-based models’ response surfaces becomes more evident (Figure 4 and Appendix A). Here, a clear optimum is seen for the CCC design-based models (Figure 4A), while the BBD-based models show a ridge system with a saddle-like nature in four dimensions, having no apparent (unique) optimum. This is reflected by the parallel contour lines in the plot (Figure 4B).

#### 2.2.3. Design Adjustment for the BDD

Despite the BBD-based models appearing to be more adequate than the CCC-based models in the statistical assessment, the saddle-like nature of the ridge system implied the possible absence of an optimum or an optimum—when, in fact, present–outside the explored design space, suggesting that the BBD might not be useful for identifying a true optimum in this application. However, such behavior of a response surface could be due to a poor choice in variable ranges [62]. Expansion of the ranges in the BBD was made (Appendix A), as perhaps an optimum in the delivery yield could yet be identified using a design that resulted in models being more statistically adequate in comparison to the CCC design.

Models fitted to the data of the BBD with extended ranges (Appendix A) indeed gave more significant model terms with a higher pred. R^2^ and other metrics match those of the original BBD-based models. Nonetheless, the PDNP size was again the only consistent significant estimate in its linear and quadratic terms, with other terms being more variable in their significance. Breaches of the underlying assumptions were not detected.

Analysis of the stationary points for each of the replicates’ models (Appendix A) proved that the saddle point-containing ridge system was resolved. A clear optimum was identified in all cases (Figure 4C and Appendix A). Moreover, the identified optimum was very similar for every replicate and corresponded well to the optimum identified by the CCC-design-based models, with only a considerable deviation for the PDNP concentration, which had now increased. Some variability in the identified optimal parameter values was still present, as reflected by the C.V. (Appendix A), but they now approached those of the CCC design-based models. Importantly, the replicates optimal conditions are geometrically close in the design space and are quasi-optimal when considered in each of the individual models. Thus, extending the variable ranges of the BBD led to models with a clear set of optimal parameter values, and that provided better basic statistical metrics regarding the model fit with respect to the CCC design-based models (Appendix A).

#### 2.2.4. Model Validation

Following the response surface analyses, it was evaluated to which extent the models can predict the delivery yield when parameter values are used which were not included in the initial designs. Therefore, twenty confirmation runs for the CCC design and twenty-two for the (revised) BBD were performed in biological triplicates, after which the mean and standard deviation were calculated for the three replicates of each run factor level combination. By averaging, the predictions of each parameter value combination for the three model replicates and calculating the standard deviation, an estimate of the models’ variability in prediction was obtained.

The results (Table 4, Appendix A) revealed that only three of the twenty means observed values significantly departed from the mean predicted value (85% accuracy) for the CCC design-based model, whereas only two of the twenty-two mean observed values did so for the BBD-based model (91% accuracy) and three of the twenty-two when extending the variable ranges of the BBD (86% accuracy). Relatively low standard deviations of both the observed and predicted delivery yields support that these calculations are not the result of large sample variability. Despite different metrics (e.g., multiple R^2^, RMSE and MAE) indicating a clear decrease in model performance when evaluating their fit to the confirmation run data (Appendix A), these accuracies suggest that the model output of each of the used designs is capable of predicting the yield within the design space with sufficient reliability. No discernable differences were observed between the CCC design-based and BBD-based models–original or extended–regarding their predictive power.

Figure 5 further illustrates the quality of the model predictions by visualizing the experimentally obtained delivery yields corresponding to several parameter value combinations across the design space. It is clearly shown that parameter value combinations in the relative vicinity of the predicted optima—e.g., 500 nm PDNPs at a concentration of 3.5 × 10^8^ PDNPs/mL—resulted in considerably higher, and even close to optimal, yields than when suboptimal parameter value combinations—e.g., 300 nm or 700 nm PDNPs, regardless of concentration and fluence—were used. Thus, the models’ identified optimum is successfully corroborated. Further support for this assertion was found in run four for all designs (Table 4, Appendix A), with a quasi-optimal factor level combination identified by the models, as they resulted in the practically highest obtained delivery yields. Moreover, it became evident that the PDNP size was the main contributor to the delivery yield, whereas the PDNP concentration and laser pulse fluence only demonstrated comparatively minor effects within the explored design space, exactly as predicted.

## 3. Discussion

Given its successful application to a wide variety of fields (e.g., microbial cultures, synthetic biology and manufacturing) [60,70,71], we have explored here the utility of RSM for the optimization of photoporation parameters. As a case study, we measured the delivery yield (i.e., a fraction of living positive cells) for FITC-dextran 500 kDa (FD500) delivery in RAW264.7 murine macrophage-like cells. RAW264.7 macrophage-like cells are a useful case study since they are of interest in fundamental and applied research regarding immunology and pathology [59], where intracellular delivery is crucial when trying to harness macrophages’ natural functions (e.g., engineering chimeric antigen receptor-macrophages) [72]. FD500 was selected as a model cargo molecule as its hydrodynamic diameter of 31 nm is close to that of relevant biological effector molecules, such as transcription factors or gene-editing nucleases [73].

Part of investigating the suitability of models fitted to the data collected using these designs is the statistical assessment of the adherence to the underlying model assumptions, as any violation can hamper the inferences made [74]. Supported by several plots, and as indicated by the non-significant results of the Shapiro-Wilk normality test, Bruesch-Pagan heteroscedasticity test and Durbin-Watson test for autocorrelation, we concluded that the model assumptions were respected when using the CCC design and BBD. The suitability of both designs is further reinforced by the absence of lack-of-fit and the significance of the fitted models in all replicates. However, a comparison of the metrics regarding the adequacy of the models did reveal some differences between both designs. Consistently higher R^2^ values, whether multiple or adjusted and lower values for the RMSE and MAE favor the choice for the BBD in this specific area [62,75,76,77].

Since the predicted optimum rarely corresponds to one of the experimental data points in the designs, the models’ predictive power was investigated. A preliminary study of the pred. R^2^ metric for the models resulting from the two designs revealed a severe decrease (>0.20) in comparison to the multiple and adj. R^2^, suggesting the low predictive power of the models [78]. Confirmation runs corroborated this finding by also establishing a true sizeable reduction in the multiple R^2^ using their data points—i.e., a measure of fit between the originally fitted curve and newly collected data points. Possible reasons for this are the presence of irrelevant model terms (i.e., overfitting) [78], model biases due to factor level errors [79] and the possibility that a second-order model does not adequately describe the relationship between the response and factors [80]. Broad design regions—e.g., by ±α-values in the CCC design—can also contribute to this issue [81]. Interestingly, the similarity of the multiple R^2^, RMSE and MAE for the confirmatory run replicates of the two designs indicated that they had equal predictive power based on statistical metrics, even though the BBD-based models had a considerable advantage in the first adequacy check. It must, however, be considered that some of the confirmatory runs for the BBD-based models were located outside its design space, thus requiring extrapolation, which is inherently more unreliable [82]. Despite the suggested lackluster predictive capabilities of both designs’ models by several calculated metrics, the high obtained accuracies based on experimental validation data points provided a first indication that the models can indeed reliably predict experimental regions of interest to some extent [83]. No difference between the outputs of the designs could be made in this aspect.

Ultimately, more important than the statistical assessment and validation of the models is their performance in practice—i.e., the ability to correctly identify the parameter values for which an optimal yield is obtained. We found that only the CCC-based models identified a true optimum in all cases, whereas the BBD-based models rather returned a saddle point-containing ridge system for two of the three replicates. The latter is an indication that certain variables—or their interactions–had an insignificant effect on the response [66], which can be caused by factor ranges being set too small [62]. Given that BBD-based models might be more suitable in some practical applications of photoporation with less flexibility, we investigated whether expanding the parameter ranges in the BBD was helpful in obtaining a true optimum. Indeed, extending the PDNP concentration and laser fluence ranges of the BBD resulted in models that contained a true optimum within the design space, which was similar to the optimum as predicted by the CCC design. Only in the predicted optimal PDNP concentration did the two designs lead to different results, with the extended range BBD-based models marking ca. 3.5 × 10^8^ NPs/mL as optimal instead of ca. 2.5 × 10^8^ NPs/mL by the CCC design-based models. In practice, however, these are rather small differences that are likely close to optimal. A comparison of the delivery yields of predicted near-optimal parameter value combinations to predicted suboptimal combinations did indeed confirm that optimum identification was effective for the models resulting from both designs. This further strengthened the preceding model validation based on confirmation runs. Considering that both designs resulted in models that were equally successful in identifying an optimum and the aforementioned effective practical validation, it is shown that (better) general statistical metrics do not necessarily lead to better/proper predictive capabilities in practice, possibly warranting some practical liberties. Additionally, the similarity of the discerned optimal conditions between the different design replicates proved that only one designed experiment is sufficient, confirming one of the key concepts in RSM. Lastly, a note on the design selection has to be made, however. Even though both designs were successfully used for optimization purposes in this study, the BBD should only be selected if the experiment permits little flexibility, as a CCC design has better predictive capacities across the entire design space [62]. Carefully made considerations of the experimental design remain crucial.

In addition to the identification of an optimum, the model output can also be used to gather information about the factors that have the most influential impact on the response. This way, a more biological/physical and mechanistic interpretation of the process can be obtained, though this necessitates a critical view of the used experimental design–especially the possibility of confounding factors and set factor ranges (in original units) [62,84]. We found that the PDNP size was consistently the most influential factor. This could be attributed to the fact that laser pulse fluences and PDNP concentrations were only varied over relatively small ranges in this study, making PDNP size the predominantly influential factor. Laser fluence was varied from 0.7 to 1.3 J/cm^2^, for which the balance of heating vs. VNB formation is fairly similar per PDNP size (Figure 3). It is, therefore, understandable that it had little influence on the final delivery yield. For a given fluence, however, there is a big difference in the predominant photothermal mechanism depending on the PDNP size. For the smaller PDNPs, heating is more prevalent, while VNB formation will be more likely for the larger PDNP sizes. Within the tested concentration range, permeabilization by VNBs appears to be more beneficial than by heating, an observation that was made before for AuNPs [19]. The fact that 500 nm PDNPs come out as more beneficial than larger ones, which also form VNBs, is likely related to the fact that they may give rise to larger and more powerful VNBs, which can lead to more damage to cells [20]. Indeed, delivery yield is the balance between delivery efficiency and cell viability. Within the tested concentration range, PDNPs larger than 500 nm inflict damage to cells too extensively (Appendix A), which negatively impacts delivery yield. It cannot be excluded that higher (still sub-optimal) delivery yields are possible for those large PDNPs if the concentration range is expanded to include lower concentrations so as to reduce toxic effects. Similarly, for higher PDNP concentrations, it may be that better, but perhaps still sub-optimal, yields are possible for the smaller PDNPs. This remains to be explored in future research.

Finally, it is of interest to consider the reduction in resources made possible by the RSM approach versus the commonly used OFAT approach. Based on previous reports on photoporation, OFAT requires about 50 experimental conditions—including technical and biological replicates—to determine the optimal nanoparticle concentration and laser fluence for a given nanoparticle size [22,25,26,53]. This means that for three PDNP sizes, this would amount to 150 experimental conditions in the OFAT approach. Instead, the RSM approach required only sixteen or eighteen runs, depending on the design, for the optimization of all three parameters, resulting in an over eightfold reduction in time and materials. Even when an additional experiment with extended ranges in the design is required, as was the case with the BBD in this study, there is still a five-fold reduction in resources.

Future work can explore the ability to use the CCC-design and BBD to optimize these photoporation parameters in other, more therapeutically relevant photoporation applications. Moreover, the inclusion of other parameters is to be investigated for parameters that can improve our fundamental knowledge of how cell biology affects the photoporation outcome. Previous research indicates that different cell types require different optimal parameter combinations [24,25,28,53], which could be related to different cell sizes, seeding density, morphology and/or their being (non-)adherent, providing possible parameters of significance. More so, some of these cell characteristics can be related to cell type-specific phenotypes (e.g., T-cells) [85,86], possibly allowing an even more refined elucidation of fundamental photoporation effects. Additionally, a combination of biological parameters with physical parameters (e.g., NP type, NP size, cargo molecule and laser fluence) can broaden such research. With rising numbers of included parameters, other venues of RSM might also prove to be of interest. For example, screening designs to reduce factor numbers, computer-aided designs, combinations of continuous and categorical factors, and other modeling techniques can offer elegant instruments to study these complex questions [57,62,66,84,87,88].

## 4. Materials and Methods

### 4.1. Synthesis of Polydopamine Nanoparticles and Functionalization with Bovine Serum Albumin

Based on a protocol reported by Harizaj et al. [53], adapted from Ju et al. [52], polydopamine nanoparticles of various sizes were synthesized. In brief, dopamine hydrochloride powder (Sigma-Aldrich) was dissolved in 20 mL of HyClone pure water (HyPure, Cell Culture Grade, VWR) at 50 °C. 1 M NaOH solution was then added under vigorous stirring, marking the onset of the polymerization process by turning the solution pale yellow, which developed into a dark brown over time. Depending on the desired size, the dopamine hydrochloride concentration was varied between 2 and 4 mg/mL in combination with varying the molar ratio of dopamine hydrochloride:NaOH between 1:0.6 and 1:0.9. The polymerization mixture was left to stir for approximately 7 h. Growth kinetics were monitored by measuring the hydrodynamic diameter of the particles every hour by dynamic light scattering (DLS) (Zetasizer Nano ZS, Malvern Instruments Co., Ltd., Malvern, UK). Upon reaching the desired hydrodynamic diameter, nanoparticles were washed several times with HyClone pure water, separating the PDNPs from the unreacted precursors by centrifugation. Sub-200 nm PDNPs reaction mixtures were divided over 1.5 mL Eppendorf tubes, and centrifugation was performed at 20,000 RCF for 20 min. Reaction mixtures of PDNPs larger than 200 nm were collected in 50 mL conical tubes, and the PDNPs with sizes between 200 and 400 nm were centrifuged for 20 min at 4000 RCF, whereas PDNPS larger than 400 nm were centrifuged for 10 min at 4000 RCF. Finally, the washed NP suspensions were sonicated at 10% amplitude for 20 s with a tip sonicator (Branson Digital Sonifier, Danbury, CT, USA) to disband agglomerates and reduce potential future NP agglomeration or even aggregation.

The colloidal stability of PDNPs in suspension and their interaction with cell membranes were improved by functionalizing the NPs with bovine serum albumin (BSA) (Biotechnology grade, VWR Chemicals, Radnor, PA, USA). Uncoated PDNP suspensions were mixed with a 20 mg/mL BSA solution in DPBS at a 1:1 volume ratio and left to be mixed vigorously overnight. Unbound BSA was removed by several washing steps with HyClone water, using the aforementioned RCFs and centrifugation times for the different PDNP sizes. At last, the resulting BSA-coated PDNPs were dispersed in HyClone water and stored at 4 °C.

### 4.2. Physicochemical and Morphological Characterization

Both the hydrodynamic diameter—reported as Z-average and referred to as the *size*—and zeta potential of the (BSA coated) PDNPs were measured using DLS (Zetasizer Nano ZS, Malvern Instruments Co., Ltd.). Nanoparticle concentrations were determined using nanoparticle tracking analysis (NTA) (NanoSight LM10, Malvern Panalytical, UK). Recording of the UV/VIS spectra was performed with a Nanodrop 2000c spectrophotometer (Thermofisher, Rockford, IL, USA). Morphological characterization and additional nanoparticle diameter measurements of the BSA-coated PDNPs were obtained by means of both scanning electron scanning microscopy (SEM) (JSM-100, JEOL, Tokyo, Japan) and transmission electron microscopy (TEM) (Tecnai G2 Spirit BioTWIN, FEI, Hillsboro, OR, USA).

### 4.3. VNB-Threshold

Vapor nanobubbles (VNBs) were generated by irradiating samples of ca. 1 × 10^9^ NPs/mL in double-distilled water (ddH_2_O), present in a 50 mm γ-irradiated glass bottom dish (MatTek Corporation, Ashland, MA, USA), with ca. 3 ns pulsed 532 nm laser light (Cobolt TorTM Series, Cobolt AB, Solna, Sweden). Lasers pulses were generated on demand using a 25 MHz pulse generator (TGP3121, Aim-TTi, Huntingdon, UK), with control over the pulse energy being provided by an adjustable DC power supply (HQ Power PS23023, Velleman Group, Gavere, Belgium). Energies were registered using an Ophir Starlite energy meter (MKS Instruments, Andover, MA, USA). The VNBs were visualized using dark field microscopy, where the increased scatter of VNBs resulted in bright white spots [19]. Short movies of this phenomenon were captured using a cMOS camera (Blackfly S GigE-Mono, FLIR, Wilsonville, OR, USA) and screen recording software, which allowed the counting of individual VNBs. The number of generated VNBs was determined in the irradiated area as a function of the applied laser fluence. A Boltzmann sigmoid curve was fitted to the data normalized against the maximal number of counted VNBs in GraphPad Prism version 8.0.0 (GraphPad Software, San Diego, CA, USA), allowing quantification of the VNB-threshold as the laser pulse fluence at which 90% of the irradiated particles generate a detectable VNB.

### 4.4. Cell Culture

RAW264.7 (American Type Culture Collection, ATCC-TIB-71) murine macrophage-like cells were cultured in DMEM/F-12 medium + GlutaMAX^TM^-I (Gibco^TM^), which was supplemented with 10% fetal bovine serum (FBS, Biowest, Nuaillé, France) and 100 U/mL Penicillin/Streptomycin (Gibco^TM^). Every 2 or 3 days, the cells reached confluency and were subsequently passaged and kept in a humidified incubator (37 °C, 5% CO_2_).

### 4.5. Photoporation for the Intracellular Delivery of FITC-Dextran 500 kDa

One day prior to performing photoporation, the RAW64.7 cells were seeded in a 96-well plate at a density of 50,000 cells/well, after which the plate was incubated overnight in an incubator (37 °C, 5% CO_2_). Next, the cells were washed once with Dulbecco’s phosphate buffered saline without Ca^2+^ and Mg^2+^ [DPBS(-/-), Biowest, Nuaillé, France] [89] after which the PDNPs, of which a series of dilutions was prepared in reduced serum culture medium [Optimized-Minimal Essential Medium (Opti-MEM), Gibco^TM^], were added to the wells. This step was immediately followed by 10 s centrifugation in a plate centrifuge (Eppendorf, Hamburg, Germany) until 1300 RCF was reached. Unbound particles were removed from the cells by washing them with DPBS(-/-). 50 µL of Opti-MEM containing 1 mg/mL FITC-dextran 500 kDa (FD500, Sigma-Aldrich, Bornem, Belgium) was then added to the wells, followed by photoporation with an in-house developed optical setup with a nanosecond laser (3 ns pulse duration, 532 nm wavelength) and equipped with a Galvano-scanner to rapidly scan the laser beam across the samples (5–6 s per well). Immediately after laser treatment, cells were washed twice with fresh culture medium to remove excess FD500 and thus prevent its spontaneous uptake by endocytic processes. Finally, the cells were prepared for flow cytometry analysis of the delivery efficiency.

### 4.6. Analysis of Intracellular Delivery Efficiency by Flow Cytometry

Delivery efficiency of FD500 was determined by washing the photoporated cells once with DPBS(-/-), adding 40 µL/well of enzyme-free cell dissociation buffer (Gibco^TM^) for 5 min on an orbital shaker (120 rpm) and then adding 110 µL of flow buffer [DPBS(-/-), 1% Bovine Serum Albumin, 0.1% Sodium Azide] containing 0.5 × 10^−6^ M of a cyanine dye monomer TO-PRO3 iodide (Invitrogen, Belgium) as cell viability dye. The samples were then measured with a CytoFLEX flow cytometry (Beckman Coulter, Krefeld, Germany). For every sample, 10,000 living cells were gated for the evaluation of the FD500 delivery in the FITC-channel by excluding TO-PRO3-positive cells in the singlet population. FlowJo™ software (Treestar Inc., Ashland, OR, USA) was used for data analysis.

### 4.7. Cell Viability Analysis

For every FD500 delivery experiment, a second plate was seeded at 50,000 cells/well of RAW264.7 cells to measure the effect of photoporation on viability. All steps were identical to the previously described protocol for flow cytometric analysis, except that FD500 was omitted from the Opti-MEM added to the cells preceding photoporation. Following laser treatment, the Opti-MEM was removed, and a fresh culture medium was added. The cells were then put in an incubator (37 °C, 5% CO_2_) for 2 to 4 h, after which the viability was assessed using the CellTiter Glo^®^ luminescent cell viability assay (Promega, Belgium). In this assay, the number of viable cells is determined by quantification of the ATP present after cell lysis, as an indicator of the metabolic activity of cells. As recommended by the manufacturer, the old culture medium was removed, and 100 µL of fresh culture medium was added, which was supplemented by an equal volume of CellTiter Glo^®^ reagent and shaken on an orbital shaker (120 rpm) for 10 min at room temperature. Next, the cell lysates were transferred to an opaque white 96-well plate (Greiner Bio-One, Belgium), and the luminescent signal was measured using a GloMax^®^ microplate reader (Promega, Belgium). Cell viability was calculated relative to the average of a technical triplicate of non-treated controls.

### 4.8. Response Surface Methodology

#### 4.8.1. Experimental Design

Three relatively well-controllable independent variables were chosen as the factors to optimize the photoporation delivery yield: PDNP size, PDNP concentration and laser fluence. The photoporation delivery yield is defined as the fraction of cells that are alive, and FD500 positive after photoporation compared to the starting number of cells and is calculated as the product of the delivery efficiency (%) as determined by flow cytometry–i.e., the percentage of FD500 + cells, after gating for viable cells–and the relative cell viability (%) as measured by the CellTiter Glo^®^ assay. Both the five-factor level Circumscribed Central Composite (CCC) design and three-factor level Box-Behnken Design (BBD) (Appendix A) were generated in R (v4.2.1) using the *rsm* package (v2.10.3) with four center-point runs (i.e., the center of design space) [67]. Randomization of the run order was performed to prevent introducing bias in the data by a time-effect. Table 5 displays the parameter values included in the design, which were selected based on prior knowledge of typical photoporation parameters, and the corresponding coded values as used in the RSM analysis, which were defined as
(1)$$x_{i}=\frac{\left(X_{i}-X_{i}^{\prime }\right)}{\Delta X_{i}}$$
where $x_{i}$ is the *i*th (dimensionless) coded factor, $X_{i}$ the corresponding *i*th factor in its original units, $X_{i}^{\prime }$ the value of the *i*th factor at the geometric center point of the design space and $\Delta X_{i}$ the difference in parameter values between coded factor level +1 and 0 for that particular factor (*i*) in its original units (e.g.,$\;\Delta X_{size}$ = 700 nm − 500 nm = 200 nm). The CCC design requires additional coded factor levels that lay out- or inside of the design space of interest (±α) to estimate the curvature of the response surface [68]. Here, an orthogonal design [66,90] was preferred over a rotatable design, setting the extra factor levels (±α) to ±1.789 [62,67]. The interested reader is referred to Myers et al. [62] and Box et al. [84] for an intuitive geometric interpretation of these designs combined with a more in-depth analysis of their characteristics.

#### 4.8.2. Model Fitting and Statistical Assessment

Once the data was collected and processed, an empirical full second-order polynomial model was fitted using ordinary least squares regression (OLS) [74,91]. The model for the three factors of interest assumes the following form
(2)$$\hat{y}={\hat{\beta }}_{0}+\overset{3}{\underset{i=1}{\sum }}{\hat{\beta }}_{i}x_{i}+\overset{2}{\underset{i=1}{\sum }}\overset{3}{\underset{j=i+1}{\sum }}{\hat{\beta }}_{ij}x_{i}x_{j}+\overset{3}{\underset{i=1}{\sum }}{\hat{\beta }}_{ii}x_{i}^{2}\;$$
in which the predicted response is denoted by $\hat{y}$, $x_{i}$ and $x_{j}$, representing the coded values of the factors, ${\hat{\beta }}_{0}$ is the estimated intercept term, the estimated coefficients of the interaction terms are given by ${\hat{\beta }}_{ij}$ and those of the quadratic terms by ${\hat{\beta }}_{ii}$. Thus, this model contains an intercept, three main effect terms ($x_{i}$), three interaction terms ($x_{i}x_{j}$) and three quadratic terms ($x_{i}^{2}$), totaling ten terms.

Model fitness was assessed using an analysis of variance (ANOVA) and a collection of summary statistics. The statistical significance of the model parameters was determined using *t*-tests, while the models’ significance and lack-of-fit were evaluated using F-tests [62]. The accuracy of the models was evaluated and compared by calculation of the coefficients of determination (multiple R^2^), adjusted coefficients of determination (adj. R^2^), predicted R^2^ (pred. R^2^), predicted residual sum of squares (PRESS), root mean squared error (RMSE) and mean absolute error (MAE) [62,75,76,92]. The normal distribution of the residuals was checked by calculating the residual mean, plotting the relation between the theoretical normal quantiles and the ordered normalized residuals (Q-Q plot) and the Shapiro-Wilk test of normality [62,93]. Homoscedasticity was tested using the Bruesch-Pagan test [94]. Lastly, the introduction of bias into the model by a time dependency was checked by the Durbin-Watson test and plotting the residuals as a function of the run order [62,95]. For all statistical tests, the null hypothesis was rejected when the *p*-value < 0.05.

#### 4.8.3. Canonical Analysis

An analytical approach, termed *canonical analysis*, was used to identify the stationary point of the fitted model and to clarify if this corresponds to a maximum, minimum, saddle point and/or ridge system. Though a general overview of the methodology is given here, the works of Myers et al. [62] and Buyske and Trout [96] excellently explain and derive all of the relevant results for canonical analysis. In brief, the aforementioned full quadratic model is represented in its matrix form
(3)$$\hat{y}={\hat{\beta }}_{0}+x^{\prime }\hat{\beta }+x\prime \hat{B}x$$
where $x^{\prime }=\left[x_{1},x_{2},\;\ldots ,\;x_{k}\right]$ represents the transpose of the vector of the factors, $\hat{\beta }\prime =\left[{\hat{\beta }}_{1},\;{\hat{\beta }}_{2},\ldots ,\;{\hat{\beta }}_{k}\right]$ the transpose of the linear coefficient estimates vector and $\hat{B}$ the k×k symmetric matrix of the estimates of the second-order coefficients
(4)$$\hat{B}=\left[\begin{array}{ccc} \begin{array}{cc} {\hat{\beta }}_{11} & {\hat{\beta }}_{12}/2 \end{array} & \cdots & {\hat{\beta }}_{1k}/2\\ \begin{array}{cc} \vdots & \ddots \end{array} & \vdots & \vdots \\ symm. & \cdots & {\hat{\beta }}_{kk} \end{array}\right]$$

By differentiating the expression in matrix form with respect to x and setting the derivative equal to 0, the location of the stationary point in vector form ($x_{s}$) can be obtained
(5)$$\frac{\delta }{\delta x}\left({\hat{\beta }}_{0}+x_{s}{}^{\prime }\hat{\beta }+x_{s}{}^{\prime }\hat{B}x_{s}\right)=\hat{\beta }+2\hat{B}x_{s}=0\overset{}{\Leftrightarrow }\;x_{s}=-\frac{{\hat{B}}^{-1}\hat{\beta }}{2}$$

Now, having identified the stationary point, it is possible to center the system at said point by performing a simple translation ($z=x-x_{s}$), thus arriving at
(6)$$\hat{y}={\hat{y}}_{s}+z\prime \hat{B}z$$

In the final step, the axes of this re-centered system are rotated in such a manner that they will become the axes of symmetry of the fitted model (i.e., principal axes). A new k×k matrix termed P is defined, whose columns are the eigenvectors associated with $\hat{B}$, together with a diagonal matrix $\Lambda =P\prime \hat{B}P$ containing the eigenvalues ($\lambda_{k}$) corresponding to those eigenvectors. The *raisons d’être* of these matrices are stated in basic linear algebra texts [97] and proven in more rigorous works dealing with spectral theory [98,99,100]. If a new vector w=P′z is introduced, the canonical form of the model is easily derived to be
(7)$$\hat{y}={\hat{y}}_{s}+\overset{k}{\underset{i=1}{\sum }}\lambda_{i}w_{i}^{2}$$

Thus, the predicted response is now purely a function of the predicted response at the stationary points, the eigenvalues of the system and the canonical variables $w_{i}$. From this polynomial, it becomes clear that, when moving away from the stationary point, the response decreases if all $\lambda_{i}<0$ and increases if all $\lambda_{i}>0$, corresponding to a maximum and minimum in the system, respectively–mixed signs of $\lambda_{i}$ indicate the presence of a saddle point, where there is no maximum nor minimum in the observed experimental region. Cases where the presence of at least one near-singular (≈0) eigenvalue in a model where its canonical form occurs signify a ridge system, where the surface curvature is small within the design space, meaning that multiple parameter value combinations result in the same (optimal) response [66]. This offers a relatively simple way to study the nature of the stationary point.

#### 4.8.4. Model Validation

Having checked the fitness of the model and having established the location of the optimum, the model was then validated through the collection of new data points and comparison of the predicted response to the experimentally obtained response. Twenty confirmation runs for the CCC design and twenty-two for the BBD were performed (*n* = 3) [101]. Twelve of the confirmation runs for the CCC design consisted of the runs for the BBD, minus the center runs, whereas fourteen of the confirmation runs for the BBD were runs used for the CCC design, also minus the center runs. Validation of one design was thus partly based on re-using data of the other design, as they both contained different coordinates in the design space. Eight additional confirmatory runs were selected in the ranges between −1 and +1 of all factors (Appendix A), according to new factor levels defined in Table 6. These eight runs were the same for both the CCC design-based and BBD-based model validation. A scheme of the used RSM procedure clarifies this strategy (Figure 6). Both the average and standard deviation of the predictions of each model and observed values from the confirmation runs were calculated. An unpaired Welch’s *t*-test was then performed between the predicted values and the observed values to check if the means were significantly different (*p* < 0.05). This allowed us to estimate the predictive capability of models fitted to the data collected according to the BBD or CCC design. A more detailed view of the performance of the individual models was obtained by calculation of the multiple R^2^, RMSE and MAE on the validation data set.

## 5. Conclusions

In this study, we successfully applied RSM to efficiently determine photoporation parameter values that result in an optimal delivery yield. A comparison of two popular designs, the CCC design and BBD, revealed equal applicability to RSM for the prediction of optimal parameter values, though the BBD requires the least flexibility. Compared to the traditional OFAT approach, RSM resulted in over eight-fold fewer experiments needed to determine the optimal conditions, with still a five-fold reduction in resources upon design revision. In future work, it will be of interest to apply the RSM framework to other cell types and cargo molecules, as well as to expand the range of parameters to gain further insight into the physicochemical (e.g., NP type, NP size, cargo molecule and laser fluence) and biological (e.g., cell density, size, type and morphology) aspects that play a role in intracellular delivery by photoporation.

## Figures and Tables

**Figure 1 ijms-24-03147-f001:**
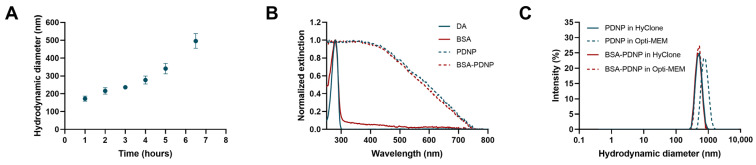
Characterization of BSA-coated and uncoated polydopamine nanoparticles (PDNPs). (**A**) Evolution of the PDNP hydrodynamic diameter in the function of reaction time as measured by DLS. All samples were sonicated by tip sonication to remove potential agglomerates. (**B**) Normalized UV/Vis spectrum of DA (blue), BSA (red), 500 nm PDNPs (dashed blue) and 500 nm BSA-coated PDNPs (dashed red). (**C**) Size distributions representing the stability of uncoated 500 nm PDNPs in water (blue) and Opti-MEM (dashed blue), showing a clear size increase after incubation in Opti-MEM. BSA-coated PDNPs maintain the same size distribution in both waters (red) and Opti-MEM (dashed red). Measured by DLS for an incubation time of 15 min.

**Figure 2 ijms-24-03147-f002:**
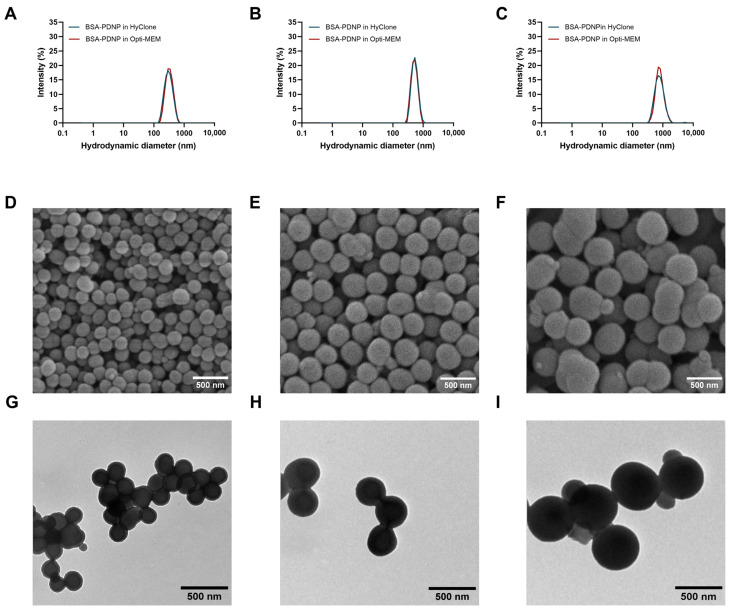
Size and morphology characterization of BSA-coated polydopamine nanoparticles (PDNPs) of various sizes. (**A**–**C**) Representative intensity size distributions as measured by DLS of BSA-coated PDNPs of 300 nm (**A**), 500 nm (**B**) and 700 nm (**C**) in water (blue) and 15 min incubated in Opti-MEM (red). (**D**–**F**) Representative SEM images of BSA-coated PDNPs of 300 nm (**D**), 500 nm (**E**) and 700 nm (**F**). Scale bar = 500 nm. (**G**–**I**) Representative TEM images of BSA-coated PDNPs of 300 nm (**G**), 500 nm (**H**) and 700 nm (**I**). Scale bar = 500 nm.

**Figure 3 ijms-24-03147-f003:**
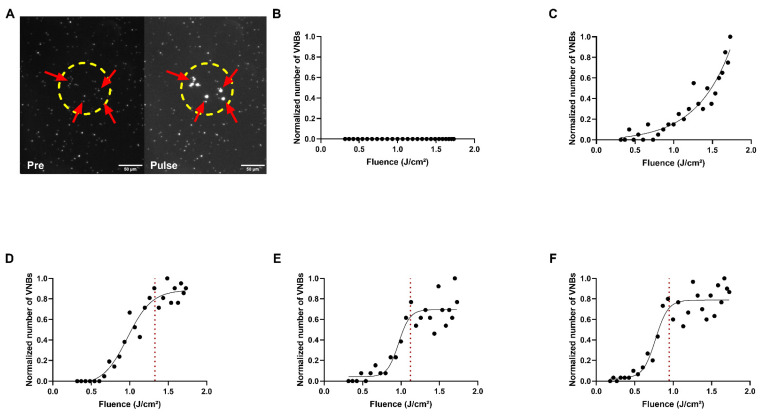
VNB formation by PDNPs. (**A**) Visualization of vapor nanobubbles (VNBs) (red arrows) by dark field microscopy before (pre) and upon (pulse) local laser irradiation with a fluence of 1 J/cm^2^ in a 500 nm PDNP dispersion (dashed yellow circle). (**B**–**F**) VNB formation as a function of laser pulse fluence for 142 nm (**B**), 300 nm (**C**), 500 nm (**D**), 700 nm (**E**) and 858 nm (**F**) PDNPs. By fitting a Boltzmann curve (solid black curve), the VNB threshold (dashed red line) could be determined for 500 nm, 700 nm and 858 nm PDNPs. Scale bar = 50 µm.

**Figure 4 ijms-24-03147-f004:**
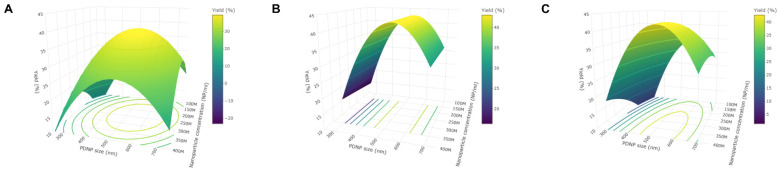
Representative response surfaces where the PDNP size was kept constant, and the other factors were varied. (**A**) CCC design-based model output visualized, showing a clear optimum. (**B**) BBD-based model output visualized, with no clear optimum being present in the investigated design space. (**C**) Revised BBD-based model output visualized, now showing a clear optimum.

**Figure 5 ijms-24-03147-f005:**
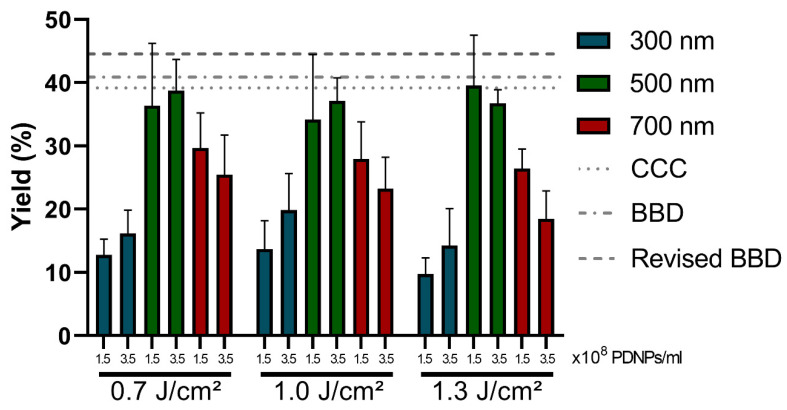
Comparison of the delivery yield experimentally obtained at different factor level combinations. Yield in the function of different PDNP sizes, PDNP concentrations and laser fluences, showing the presence of unfavorable factor level combinations. Near-optimal parameter value combinations led to delivery yields close to the mean (±standard deviation) of the predicted optimal delivery yield by the CCC design- (dotted line; 39.168 ± 2.965%) and revised BBD-based (alternating stripe and dot line; 44.577 ± 1.971%) models. In the case of the BBD, these values were near the mean of its predicted stationary point by BBD-based models (striped line; 40.885 ± 2.763%). Error bars = standard deviations.

**Figure 6 ijms-24-03147-f006:**
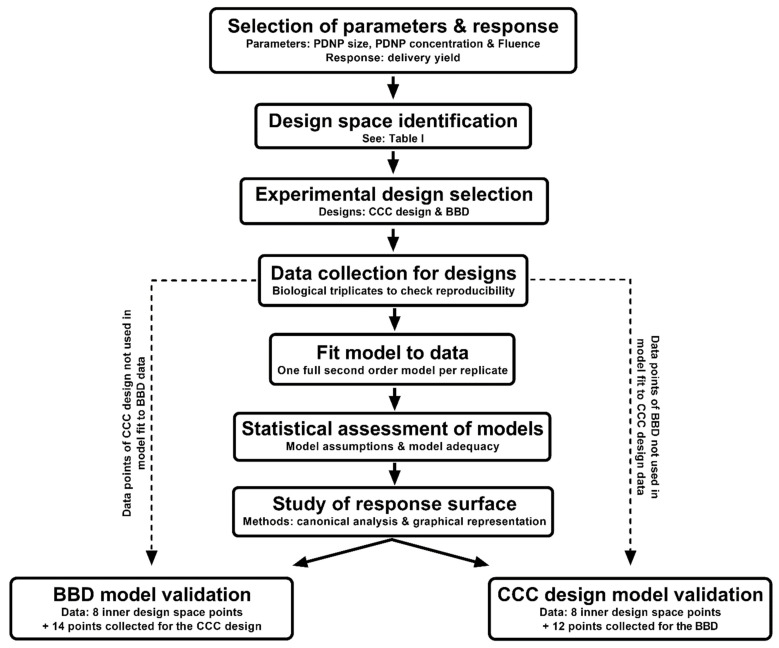
Scheme representing the complete RSM procedure used in this study and clarifying the model validation strategy.

**Table 1 ijms-24-03147-t001:** Summary of the average (±standard deviation) hydrodynamic diameter (size), polydispersity index (PDI), and zeta potential (Z.P.) of the BSA coated PDNPs with sizes corresponding to factor levels −1, 0 and 1 in both HyClone Pure water and Opti-MEM. Z.P. was only measured for BSA-coated PDNPs in HyClone Pure water.

	300 nm Factor Level: −1		500 nm Factor Level: 0		700 nm Factor Level: +1	
	HyClone	Opti-MEM	HyClone	Opti-MEM	HyClone	Opti-MEM
Size (nm)	290.8 ± 3.5	309.0 ± 6.8	516.1 ± 9.3	492.3 ± 5.8	728.4 ± 8.7	754.6 ± 23.6
PDI	0.084 ± 0.020	0.078 ± 0.015	0.051 ± 0.014	0.050 ± 0.019	0.118 ± 0.048	0.112 ± 0.043
Z.P. (mV)	−33.0 ± 0.1		−37.9 ± 0.9		−40.6 ± 0.5	

**Table 2 ijms-24-03147-t002:** Summary statistics for the optimization of photoporation yield using PDNPs as photothermal NPs and FD500 as cargo molecules, with both a CCC design and BBD. Triplicates were performed to assess model output reproducibility. The mean and standard deviation of the triplicates are presented. Values indicated in red and with * are significant at *p* < 0.05 for a one-sided unpaired Welch’s *t*-test between the CCC design- and BBD-based model summary statistics.

	CCC	BBD
Statistic	Mean ± Standard Deviation	Mean ± Standard Deviation
Multiple R^2^	0.871 ± 0.038 *	0.951 ± 0.009 *
Adjusted R^2^	0.726 ± 0.081 *	0.877 ± 0.022 *
PRESS	1921 ± 821	907 ± 79
Predicted R^2^	0.145 ± 0.275	0.416 ± 0.046
RMSE	3.957 ± 0.824 *	2.180 ± 0.191 *
MAE	3.340 ± 0.505 *	1.850 ± 0.185 *

**Table 3 ijms-24-03147-t003:** Summary of the stationary points (S.P.) of all replicates for both the CCC design and BBD in original units, delivery yield at the S.P., coefficients of variation (C.V.) and all corresponding eigenvalues, providing more information on the nature of the S.P.

	CCC				BBD			
	Replicate 1	Replicate 2	Replicate 3	C.V.	Replicate 1	Replicate 2	Replicate 3	C.V.
Size (nm)	534.956	550.250	536.688	1.549	524.159	540.399	567.548	4.029
Concentration (×10^8^ NPs/mL)	2.594	2.455	2.504	2.800	2.823	2.404	1.279	36.817
Fluence (J/cm^2^)	0.950	0.841	0.939	6.594	1.076	0.745	0.793	20.528
Delivery yield at S.P. (%) with 95% confidence intervals	41.899 ± 7.044	39.591 ± 7.684	36.015 ± 5.214	7.569	37.781 ± 3.981	43.076 ± 4.515	41.798 ± 8.796	6.758
Eigenvalues	−2.393	−2.053	−2.718	N.A.	−1.491	0.693	0.915	N.A.
	−3.822	−4.777	−4.169		−3.494	−0.956	−0.851	
	−10.576	−10.385	−10.025		−16.884	−18.166	−17.735	

**Table 4 ijms-24-03147-t004:** Summary of the unpaired Welch’s *t*-test between the averaged predicted values and averaged observed values for the BBD-based models, with standard deviations (SD) also reported. Values indicated in red and with * are significant at *p* < 0.05.

	Variables			Observed Values		Predicted Values		*t*-Test	
Run	Size	Concentration	Fluence	Mean	SD	Mean	SD	*t*-Value	*p*-Value
1	0.25	0.5	0.5	29.437	1.902	38.215	1.973	−5.548	0.005 *
2	−0.5	−0.5	0.5	24.316	7.639	33.033	4.837	−1.670	0.183
3	−0.5	0.5	−0.5	24.166	5.488	33.613	3.379	−2.539	0.077
4	0.25	−0.5	−0.5	39.794	4.459	39.713	4.606	0.022	0.984
5	0.25	0.5	−0.5	34.062	1.541	39.812	2.952	−2.991	0.058
6	0.25	−0.5	0.5	32.018	4.620	39.429	3.919	−2.119	0.103
7	−0.5	0.5	0.5	28.528	5.513	33.866	2.278	−1.550	0.230
8	−0.5	−0.5	−0.5	20.523	3.739	31.467	4.622	−3.188	0.035 *
9	−1	1	1	19.168	7.338	20.149	2.872	−0.216	0.845
10	1	0	0	26.262	4.842	26.630	1.628	−0.125	0.910
11	−1	1	−1	19.090	7.274	18.492	7.245	0.101	0.925
12	−1	−1	1	13.305	5.476	17.069	8.981	−0.620	0.576
13	0	0	1	23.111	8.938	38.893	2.800	−2.919	0.081
14	0	0	−1	38.841	2.164	39.541	4.478	−0.244	0.824
15	1	1	1	20.239	3.217	17.903	3.677	0.828	0.455
16	1	1	−1	28.705	1.443	26.108	5.100	0.849	0.475
17	−1	−1	−1	15.425	2.595	10.159	6.440	1.314	0.292
18	−1	0	0	22.932	9.079	18.488	2.998	0.805	0.492
19	0	−1	0	34.199	10.266	38.359	6.726	−0.587	0.593
20	0	1	0	37.972	5.081	38.609	2.644	−0.193	0.860
21	1	−1	−1	34.704	3.256	28.689	7.757	1.238	0.313
22	1	−1	1	28.532	3.118	25.737	6.291	0.690	0.541

**Table 5 ijms-24-03147-t005:** Overview of the parameter values included in the design and their translation to coded values (bold) as they are used in the RSM model.

	Parameter Values				
Size (nm)	142	300	500	700	858
Concentration (NPs/mL)	0.789 × 10^8^	1.5 × 10^8^	2.5 × 10^8^	3.5 × 10^8^	4.289 × 10^8^
Fluence (J/cm^2^)	0.463	0.7	1	1.3	1.537
**Coded factor levels**	**−1.789**	**−1**	**0**	**+1**	**+1.789**

**Table 6 ijms-24-03147-t006:** Overview of the parameter values included in the confirmatory runs and their conversion to coded values (bold) as they are used in the RSM model. * Coded factor level for a size of 550 nm is 0.25, while the other parameter values in that column correspond to coded factor levels of 0.5.

	Parameter Values	
Size (nm)	400	550
Concentration (NPs/mL)	2.0 × 10^8^	3.0 × 10^8^
Fluence (J/cm^2^)	0.85	1.15
**Coded factor levels**	**−0.5**	**0.25 or 0.5 ***

## Data Availability

The data presented in this study are available on request from the corresponding author.

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
