# Peer review of "Response Surface Methodology to Efficiently Optimize Intracellular Delivery by Photoporation"

_ijms, 2023, doi:10.3390/ijms24043147_

Round 1
Reviewer 1 Report
This article presents a study focused on the development of a response surface methodology to efficiently optimize intracellular release by photoporation based on the use of 500kDa FITC-dextran molecules, investigating the effect of reagent concentration and size on optimization. of process. The paper is well-written and sounds interesting. However, it is necessary to address some aspects to improve the quality of the manuscript before paper publication.
-A similar study has been published by Kevin Braekmans et al. in Nanoscale (2021) 13, 6592-6604, doi. 10.1039/D0NR05067A exploring AuNP and MagP nanoparticles for photoporation purposes. It is worth highlighting the novelty of the article as well as the differences with respect to the previous one in the introduction section.
-No information is reported on the stability of these systems. It should be very important to evaluate this aspect by exploring the behavior of SPR or UV-visible spectra over time, to rule out possible aggregation phenomena.
-Please provide TEM or AFM images for the synthesized nanosystems to better characterize their size and morphology.
-The concentrations of the reactant used for viability assay must be clarified in the method section of the manuscript.
Author Response
This article presents a study focused on the development of a response surface methodology to efficiently optimize intracellular release by photoporation based on the use of 500kDa FITC-dextran molecules, investigating the effect of reagent concentration and size on optimization. of process. The paper is well-written and sounds interesting. However, it is necessary to address some aspects to improve the quality of the manuscript before paper publication.
We thank the reviewer for appreciating our work and for providing valuable suggestions on how to further improve our manuscript. We have carefully considered all the comments and given a detailed response for each question below. We also indicate how the manuscript was adjusted accordingly.
- A similar study has been published by Kevin Braekmans et al. in Nanoscale (2021) 13, 6592-6604, doi. 10.1039/D0NR05067A exploring AuNP and MagP nanoparticles for photoporation purposes. It is worth highlighting the novelty of the article as well as the differences with respect to the previous one in the introduction section.
We would like to thank the reviewer for this comment. It seems that there may have been a misunderstanding about the purpose of this work. As explained in lines 68-70, previous photoporation studies – among which the mentioned paper which is now cited – only used the traditional One-Factor-at-A-Time (OFAT) approach. Lines 78-80 then explain that the present study takes the next step by using RSM in the optimization of the photoporation process instead of relying on OFAT. This is where the novelty of our present study lies.
- No information is reported on the stability of these systems. It should be very important to evaluate this aspect by exploring the behavior of SPR or UV-visible spectra over time, to rule out possible aggregation phenomena.
With the polydopamine nanoparticles being of a polymeric nature and having no plasmonic characteristics, the use of both SPR and UV/Vis spectra to check for stability/potential aggregation in function of time is not an option to our knowledge. However, we have now included DLS size and PDI measurements of these particles 90 days after synthesis in the Supplementary Information (Table SIV) and have referred to it in the main manuscript (Lines 344-346). Considering that the hydrodynamic size and PDI is virtually identical on day 1 and day 90, we can conclude that nanoparticle aggregation or agglomeration is not taking place in function of time over a time span of at least 3 months. This corroborates earlier findings on the stability of these nanoparticles in HyClone Pure water and Opti-MEM culture medium, showing no signs of aggregation over several months (cfr. Harizaj et al., Adv. Funct. Mater., 2021, Supporting information, Figure S3).
- Please provide TEM or AFM images for the synthesized nanosystems to better characterize their size and morphology.
Thank you for this relevant suggestion. Both TEM and SEM were performed on all used nanoparticle sizes to characterize the general morphology and nanoparticle diameter. The data has now been added to the main manuscript (Figure 3, Lines 348-354) and Supplementary Information (Table SV & Figure S2). While the SEM/TEM sizes are somewhat smaller than the hydrodynamic diameter measured by DLS, as is usually the case, the relative increase in size between particles remains unchanged. Most importantly, all particles are quite uniform and have a near spherical morphology.
- The concentrations of the reactant used for viability assay must be clarified in the method section of the manuscript.
We have now provided the used volume of the reagent in the relevant section of Methods according to the reviewer’s suggestion. However, no concentrations can be provided as the composition of the reagent is proprietary to Promega and not publicly available. To avoid confusion for the reader, in line 193 we have clarified this by referring to the manufacturer’s protocol.

Reviewer 2 Report
The authors presented an interesting study that supported the data with calculations using statistical methods. In general, the article is large, informative and creates a good impression. However, there are a number of issues that require correction of the manuscript (they are listed below). In addition, it is necessary to check the text for omissions and typos.
1.Line 126 – the abbreviation ddH2O was not introduced.
2.Lines 137-140 – the program used for this purpose should be indicated.
3.Line 152 – the composition of DPBS(-/-) is not shown.
4.Line 153 – what is “Opti-MEM”? (solution/budffer/medium)/ Line 168 - TO-PRO3 (the same). Non-traditional abbreviations should be written in full first time with abbreviation in bracket. Otherwise it is not clear for specialists from other area.
5.Section 2.8. All the discussion and explanation of chosen method should be moved to the results and discussion. Section 2.8. should contain methodology only without explanation and cocnlusions.
6.Fig. 2. The figure consists of 8 ones and thus legends are small and partially unreadable. My recommendation is to divide it into two or transfer part за figure to the supplement information for better perception and reading.
7.Table IV is too big. It might be better to place it in the Supplement Information.
Author Response
The authors presented an interesting study that supported the data with calculations using statistical methods. In general, the article is large, informative and creates a good impression. However, there are a number of issues that require correction of the manuscript (they are listed below). In addition, it is necessary to check the text for omissions and typos.
We thank the reviewer for appreciating our work and providing valuable comments that help to further improve our manuscript. We have carefully considered all your concerns and provided a detailed response per each question below.
- Line 126 – the abbreviation ddH2O was not introduced.
Thank you for pointing out this oversight, the text has now been corrected by adding the full description of ddH2O.
- Lines 137-140 – the program used for this purpose should be indicated.
We have adapted this section of Methods according to the reviewer’s suggestion.
- Line 152 – the composition of DPBS(-/-) is not shown.
We have adapted this section of Methods according to the reviewer’s suggestion to clarify both the name, the fact that this buffer was purchased from a manufacturer (Biowest) and which publication describes the formulation.
- Line 153 – what is “Opti-MEM”? (solution/buffer/medium)/ Line 168 - TO-PRO3 (the same). Non-traditional abbreviations should be written in full first time with abbreviation in bracket. Otherwise it is not clear for specialists from other area.
We have clarified the nature of these substances in the Methods section according to the reviewer’s suggestion. However, it has to be noted that the TO-PRO3 name is a product name and not a (logical) abbreviation of the compound. Thus it cannot be written in full. Likely, it is derived from the IUPAC/chemical name of the compound 3-methyl-2-((E)-3-((E)-1-(3-(trimethylammonio)propyl)quinolin-4(1H)-ylidene)prop-1-en-1-yl)benzo[d]thiazol-3-ium iodide.
- Section 2.8. All the discussion and explanation of chosen method should be moved to the results and discussion. Section 2.8. should contain methodology only without explanation and cocnlusions.
Following the reviewer’s request, we have now removed some parts in section 2.8. Nonetheless, as the target audience of the International Journal of Molecular Sciences might not be familiar with several of the used methods, we do feel it is warranted to explain some of the key aspects of the methodology (e.g., design orthogonality, design rotatability & canonical analysis) that are not commonly found in IJMS publications. This way, we try to ensure that a good understanding of the obtained results is possible without requiring readers unfamiliar with RSM to delve into the literature. We try to reach as broad an audience as possible and hope our approach allows for better accessibility and a less daunting threshold to incorporate RSM in other research.
- The figure consists of 8 ones and thus legends are small and partially unreadable. My recommendation is to divide it into two or transfer part за figure to the supplement information for better perception and reading.
To improve readability, we have now split up Figure 2 in Figure 2, Figure S1 & Figure 3.
- Table IV is too big. It might be better to place it in the Supplement Information.
As suggested by the reviewer, we have now moved Table IV to the Supplementary Information (Table SVI).
Reviewer 3 Report
In the present study, the authors employed Response surface methodology to efficiently optimize intracellular delivery by photoporation. The experiments in this study are well-designed, and the results are well-presented and discussed. Overall, this work is of interest and the results are promising. Therefore, the present manuscript is strongly recommended for publication in this journal. However, the reviewer recommends surface morphology––transmission electron microscopy––studies be added to the future studies section in the manuscript. Moreover, it would be better to add polydispersity index values with particle size analysis data.
Author Response
In the present study, the authors employed Response surface methodology to efficiently optimize intracellular delivery by photoporation. The experiments in this study are well-designed, and the results are well-presented and discussed. Overall, this work is of interest and the results are promising. Therefore, the present manuscript is strongly recommended for publication in this journal. However, the reviewer recommends surface morphology––transmission electron microscopy––studies be added to the future studies section in the manuscript. Moreover, it would be better to add polydispersity index values with particle size analysis data.
We thank the reviewer for appreciating our work. A more complete characterization of the BSA coated polydopamine nanoparticles has now been included by adding both TEM and SEM images of several relevant nanoparticle sizes (Figure 3). Regarding the inclusion of the polydispersity indices (PDIs) for the nanoparticle suspensions, we would kindly refer to Table III & Table SIII of the original manuscript¸ where the PDIs are shown already. In the revised manuscript, the newly added Table SIV also includes the PDIs next to the aforementioned tables.

Round 2
Reviewer 2 Report
The article is improved by authors, it can be accepted